# Fabulation, Machine Agents, and Spiritually Authorizing Encounters

J. Loewen-Colón [1,*] and Sharday C. Mosurinjohn [2,*]

1   School of Computing, Queen's University, Kingston, ON K7L 3N6, Canada
2   School of Religion, Queen's University, Kingston, ON K7L 3N6, Canada
*   Correspondence: jbloewen@syr.edu (J.L.-C.); sharday.mosurinjohn@queensu.ca (S.C.M.)

**Abstract:** This paper uses a Tavesian model of religious experience to make a modest theorization about the role of "fabulation", an embodied and affective process, to understand how some contemporary AI and robotics designers and users consider encounters with these technologies to be spiritually "authorizing". By "fabulation", we mean the Bergsonian concept of an evolved capacity that allows humans to see the potentialities of complex action within another object—in other words, an interior agential image, or "soul"; and by "authorizing", we mean "deemed as having some claim to arbitration, persuasion, and legitimacy" such that the user might make choices that affect their life or others in accordance with the AI or might have their spiritual needs met. We considered two case studies where this agency took on a spiritual or religious valence when contextualized as such for the user: a robotic Buddhist priest known as Mindar, and a chatbot called The Spirituality Chatbot. We show how understanding perceptions of AI or robots as being spiritual or religious in a way that authorizes behavioral changes requires understanding tendencies of the human body more so than it does any metaphysical nature of the technology itself.

**Keywords:** fabulation; attribution and ascription; machine agents; spiritually authorizing encounter; Spirituality Chatbot; ELIZA effect; Mindar

## 1. Introduction

This paper makes a modest but necessary contribution to theorizing machine agency from a religious studies perspective, using the Bergsonian concept of "fabulation", The concept of fabulation acts as a model of how humans attribute agency to things which we reframe in light of the Tavesian building-block approach to how humans pick out certain things as special ("singularization") and attribute causal power to them ("simple ascription"). The role of fabulation, an embodied and affective process, and the building-block framework are useful for understanding how some contemporary AI and robotics designers and users consider encounters with these technologies to be spiritually "authorizing". To be clear, we do not claim that machine agents[1] of any kind that currently exist have attained the ontological status of human personhood via their roles as "spiritual authorizers". By "authorizing", we mean "deemed as having some claim to arbitration, persuasion, and legitimacy" such that the user might make choices that affect their life or others in accordance with the AI or might have their spiritual needs met. This could mean accepting that the AI has a legitimate role in a particular religious ontology and trusting its metaphysical conceptualization of the user's needs in spiritual care contexts. It might also mean the AI could help the user foster affective intimacy with whatever is considered divine or mediate divine presence (i.e., reproducing the authorizing function of monks, nuns, priests, gurus, imams, rabbis, etc.).

While much of the scholarly discussion of AI and "spiritual authority" has focused on theological debates over personhood and, therefore, its candidacy as a religious leader (Straine 2020), we do not have any theological stake in emic debates about how AI disrupts

existing community authority structures or complicates doctrinal authority. Other strains of this scholarly discussion have focused on extreme examples of fear, rejection, or obsession with AI, as with those who see the technological singularity as apocalyptic—either a revelation or an extinction event.[2] Though public concern needs to be consistently addressed, specifically about the social role of AI, the research approach to these attitudes has settled on explaining them via the uncanny in human–machine interaction (Weisman and Peña 2021), or by exploring the role of science fiction in shaping cultural perspectives toward machine agents (Straine 2020). Beth Singler recently summed this up in a BBC video, in which she said, "Sometimes machines can give the wrong answer. Sometimes AI can be artificially stupid rather than artificially intelligent". And regarding the question of religion and AI specifically, she commented, "I don't necessarily think that people are having real religious experiences when they say they feel blessed by the algorithm, but it's a way of talking, a metaphor, that inherently gives AI supremacy over their lives and their choices" (Betizza 2021).

## 2. The Problem: How Should We Understand Encounters Deemed "Spiritual" or "Religious" with Machine Agents?

Our concern is precisely with what happens when people engage these metaphors that give AI "supremacy", or when an encounter with AI is deemed spiritually "authorizing", or legitimate, and why. We are not interested in the "real"-ness of these experiences in terms of ontotheology, nor are we interested in doing a discursive or ideology critique of this language. Instead, we make a Tavesian move to look at the materiality of *experiences deemed religious*—the bottom-up question of what is happening in our physiology, our embodied relations, and our affective atmospheres (Taves 2020). To that end, we also prefer the second-order framing of an *encounter* deemed spiritual or religious. We find the differentiation between encounter and experience important because we are not looking to extreme examples of sensory overload or mystic revelation, but instead, those events and experiences that are relatively normative *in* a religious context but considered "special" *as* religion (Taves 2010).[3] We are interested in why encounters with machine agents such as a robotic Buddhist priest or a "spiritual chatbot" actually feel like a typical religious encounter with human agents. Understanding how these encounters come to be considered as *ordinary* special ones also involves paying attention to the materiality of machine agents and the aesthetics and affects of the digital physical things we interact with when we interact with AI.[4]

To that end, we will examine and apply a fabulative reading of AI to two machine agents referenced above: the robotic Buddhist priest Mindar at the Kodaji Temple in Kyoto, Japan; and a software program called "The Spirituality Chatbot" being developed by information studies scholar Lu Xiao at Syracuse University. While the builders of Mindar hope to equip it with a higher-functioning AI to give "the robot some measure of autonomy" (Holley 2019), it currently "just recites the same preprogrammed sermon about the Heart Sutra over and over" (Samuel 2020). However, Mindar does have other two-way features, or receptive sensitivities, such as a tiny video camera in the robot's left eye, which it uses to surveil the room for the presence of an audience. In contrast, the Spirituality Chatbot uses a low-level AI in the form of an algorithm using the Juji program of question–answer pairs built from Reddit discussion data based on the Bible or Qur'an. When a user types in a question, the Chatbot can organize a response from this corpus.

In the media and literature on Mindar and the Spirituality Chatbot, we see that users themselves deemed their encounters with the robot and the Chatbot as spiritually authorizing. So, how does this happen? What is it about humans that we are willing to take machine agents seriously as religious authorities? Previous explanations have turned to concepts like the "ELIZA effect", which is the tendency of humans to perceive their encounters with AI as more like encounters with humans than not. While the ELIZA effect captures something vital about the psychology of human–machine interactions, it does not capture enough complexity to provide satisfying explanatory power for the nature of

human–machine encounters. We suggest that a better explanation requires understanding the embodied capacities of the human users who encounter these agents.

Thus, we theorized an answer to these questions using recent studies from the phenomenology of VR and psychology of religion to explain that user encounters with AI specifically (as in the Spirituality Chatbot) and machine agents broadly (as in Mindar) rely on a series of embodied human tendencies that contribute to a sense of self-reflexive agency that informs and is informed by perception. We clustered these tendencies under Henri Bergson's religious concept of fabulation, which plays a significant role in formulating and experiencing agency in digital and nondigital environments. In other words, we concluded that perceiving machine agents as intelligent, agential, or empathetic, and, consequently, spiritually authorizing, has as much to do with the user's own agency as it does the actuality of the algorithmic functions.

### 3. The ELIZA Effect: An Authorizing Encounter with Perceived Agency

When it comes to digital technology, the human tendency to perceive agency is best exemplified and often explained using the story of Joseph Weizenbaum's computer program ELIZA. Weizenbaum developed ELIZA as a simple computer program that would analyze text input from users, take apart the sentences and organize them by syntax, and then transform and export a response based on a database of language structures and words. The program itself consisted of multiple different "scripts", of which the most famous simulated the psychotherapy of Carl Rogers. By taking on linguistic roles, ELIZA responded in ways that felt agentially plausible to those who encountered it. Weizenbaum described the algorithm as "a set of rules rather like those that might be given to an actor who uses them to improvise around a certain scene" (Weizenbaum 1976, p. 3). As D. Fox Harrell (2019) explained: "Weizenbaum's aim in creating Eliza was not to show that such a program was 'intelligent', and hence an AI system, but rather to show a clever approach to text manipulation and (later) to consider how this revealed something about the psychological traits of *humans*". This tendency and trait of humans registering their encounters with AI as more like humans than not, or ascribing them with agency and intentionality, was dubbed by the scholar of cognitive science and comparative literature Douglas Hofstadter as the "ELIZA effect".

The ELIZA effect is an example of the embodied human capacity for agent detection. In recent years, the study of agency detection in externalized objects and environments has expanded into the fields of cognition, neuroscience, and computer science. According to cognitive scientist Marc Andersen, the basic concept is that "humans have evolved a perceptual apparatus that is hard-wired to be overly sensitive to the detection of agents". (Andersen 2019, p. 65). However, what is often missed in studies of agency detection is that the same tool is actually applied not only to externalized agents, but also to our own agency. Our sense of self-as-agent is shared with our capacity to see agency in other things. This is best exemplified in Henri Bergson's theoretical psycho-physio-religious concept of "fabulation", and further unpacked with Taves' theory of attribution and ascription in the making of special things.

### 4. Agency Detection: Henri Bergson's Fabulation and Ann Taves' Attribution and Ascription

*4.1. Bergson*

The philosopher Henri Bergson ([1932] 1986) built the concept of fabulation in his work *The Two Sources of Morality and Religion*. In short, Bergson conceived fabulation as an evolutionary response to counteract the faulty aspects of human intelligence. According to Bergson, as humans became more intelligent, they began to recognize their own individuality more clearly, and, consequently, their own individual death. This recognition, he argued, encouraged the individual toward selfishness and resistance to society, as the intelligence counsels "egoism first" (Bergson [1932] 1986, p. 111). Fabulation was nature's way of counteracting the negating tendency of the intellect. Functionally, fabulation al-

lowed humans to see the potentialities of complex action within another object; that the object might also have a "soul" (with "soul" meaning, in this context, some sort of bounded, comprehensive organ of consciousness). Like agency detection, fabulation posits agential images both externally and internally (Bergson [1932] 1986, p. 123). Since humans could perceive agency in images beyond their individual body, or "souls" that could exist outside their own, the fabulative tendency allowed humans to perceive a type of persistence of the soul beyond death. It also allowed for the creative imaging of gods, which commonly appeared in the form of a deity that "prevents or forbids" (Bergson [1932] 1986, p. 112). Bergson went on to argue that the embodied tool of fabulation, when coupled with the authoritative power of religion, "succeeds in filling in the gap, already narrowed by our habitual way of looking at things, between a command of society and a law of nature", between our selfish and illusory recognition of the distinctness of our bodies from those around us and the deep, immersive connection we have to those other bodies, including machine agents (Bergson [1932] 1986, p. 5).

To reiterate, Bergson theorized the body as our primary tool for mediating our own agency and intentionality in addition to the sensations of the world outside our bodies. Fabulation, which is at the core of our tendency toward religion as social obligation, or the pressure of the collective in the face of our individual finitude, is a part of this mediating process and plays an important role, along with intuition, in our capacity for experiences deemed religious. It allows for the recognition that we exist as creatures in time beyond the present, and encourages solidarity with the people and world around us. Furthermore, fabulation is at the root of our embodied capacity to experience technologies such as VR and our human tendency to perceive the agential power of AI (Loewen 2019). Our ability to attach a sense of "self" to a digital–virtual image is only a difference in degree between recognizing the attachment of a "self" to a machine agent. When we act, move, and live in the world, we attach a sense of agency to our own image and the images we see doing the same, especially those that act novelly (even minimally). In other words, agency is not identity; it is behavior.

*4.2. Taves*

The concept of fabulation essentially specifies an aspect of Ann Taves' process of simple ascription, and Taves' physiological understanding of ascription further explains the embodied mechanisms of action of fabulation. Taves' landmark work *Religious Experience Reconsidered: A Building-Block Approach to the Study of Religion and Other Special Things* (2010) introduced simple ascription as one among four building blocks: (1) "singularization", which is the process of deeming something as special; (2) "simple ascription", which refers to when someone assigns the quality or characteristic of specialness to an individual thing; (3) "attribution", which is when people attribute causal power to a thing, as in "the goddess of knowledge caused her to give great advice"; and (4) "composite ascription", in which simple ascriptions are incorporated into more complex formations; i.e., those complex formations people tend to call "religions" or "spiritualities".

For Taves, the fullest expression of agency seems to be an equation of causality plus intentionality—that is, a being has the power to affect things, *and* they have reasons for doing what they are doing. There is a mindlike quality. If people ascribe a religious, spiritual, or otherwise special character to objects, places, events, or experiences, and they respond to such qualities, then *when* they respond, they are acting as though these qualities exist as such independent from them and have the power to passively elicit a response from them. Taves' theory posits that ideas carried up by unusual experience are actually things inside of us, such as reinterpretations of memories and interpretations of perceptual experiences, but it *feels* to people that they come from outside them, not that they have been carried up from within them, because they are coming from places in our brains and bodies to which we do not have conscious access. This basically explains how it can be that human bodies generate all kinds of experiences, but that the people having the experience may attribute it to something outside themselves.

Put together into one methodological picture, the building-block approach gives us an account of "the creation of special things through a process of singularization, in which people consciously or unconsciously ascribe special characteristics to things" and "the attribution of causality to the thing or to behaviors associated with it" (Taves 2010, p. 13). Once there are things that are thought to have both special qualities and causal powers, more elaborate cultural systems might keep building around them: these are the things people tend to call religions or spiritualities—"special paths", to use Taves' second-order term. This framework helps us to articulate that the machine agents in our case studies are enrolled in existing special paths (Buddhism for Mindar; the various respective traditions of the Spirituality Chatbot's users), but they are not in these cases occasioning special experiences that are leading to new special paths—in other words, new religious movements are not at this time springing up around Mindar itself, or the Spirituality Chatbot itself. Rather, it is the ambiguity of not yet knowing what the machine agent can do that makes them intriguing as potentially intentional as well as causal. We can experience them as potential bearers of some internal unifying apparatus of consciousness, which might fabulate like us.

### 5. Case Study: Mindar

Nestled within the famed Kodaji Temple gardens in Kyoto, Japan, the robotic Zen priest Mindar actively performs Buddhist prayers and rituals (Hardingham-Gill 2019). The silicon model of Kannon Bodhisattva (Buddhist Goddess of Mercy) stands 6 feet tall and contains roughly 132 pounds of metal wires and rubber. Mindar cost over a million dollars to develop, and while its current software is relatively basic, its programmers eventually plan on incorporating machine-learning algorithms so that the robot can provide individualized spiritual and ethical care. Mindar's cocreator, Monk Tensho Goto, says that the impetus was

> to create something as humanlike as possible—something that embodied the person who spread the word of the Buddha 2500 years ago, by entering as many Buddhist teachings as possible into a computer. But even though AI is a big word these days, it's not possible to create a human being with AI. And so we began to explore the idea of an AI Buddhist statue that could teach us—but something more than just a computer full of data! We wanted to make a Buddha that could move, smile, and look you in the eyes as it shares its teachings. (qtd. in DW Shift 2020)

Goto's comment points directly toward the common hard/soft, or weak/strong, problem of AI: whether or not these powerful algorithms we humans have created will ever gain sentience, cognition, or consciousness. But in an important sense, the problem does not matter for Mindar—people are having spiritually authorizing religious encounters with it, regardless of its personhood status, or more popularly stated, whether or not Mindar actually has a "soul". The ontological status of Mindar's consciousness, soul, or subjectivity is secondary to its capacity for triggering spiritually authorizing encounters.

For example, according to video interviews, some, perhaps even many, Buddhist visitors to Kodaji Temple accept Mindar's sermons and blessings as a legitimate part of the temple experience they normally have. One woman claimed that when she made eye contact with Mindar, "it felt like it had a soul" (qtd. in Betizza 2021). Here the woman ascribes a "soul" to Mindar, or fabulates its agency, in a way that validates her otherwise ordinary experience of praying and meditating in the temple. Another woman sees robots such as Mindar as being able to "pass on Buddhism to younger people" (qtd. in Betizza 2021). In other words, they accept Mindar as performing the same kind of mediation as the human priests are performing. They do not deify Mindar, but consider it similar to other priests and religious figures, albeit one made of silicon and aluminum. These visitors have an encounter with Mindar that is institutionally religious and, despite the basic algorithms involved, an ordinary religious experience. In other words, their experience is normative in their religion and only special as religion, in the Tavesian sense, rather than special because

the machine agent and the encounter with it are being singularized in a way that suggests a new special path.

The fact that Mindar is encountered in a Zen temple, supported by a digital–virtual display that involves images of other humans participating in the ritual experience, and accompanied by sounds and soothing music, cannot be ignored. The context and quality of the experience make the encounter "religious", but the attribution of agency is a result of the fabulative tendencies of the human users that are either enhanced or dismissed and valenced by the context of the encounter. Because Mindar is encountered in a religious temple and advertised as a spiritual agent, users are primed to think and experience Mindar's agency as "spiritual". People might assume that with Mindar, the context (in the Buddhist temple, with human priests) is the most crucial part, but context or atmosphere helps primarily with the affective valencing of the encounter, not the attribution of agency or authority. The comparatively decontextualized Spirituality Chatbot highlights the truth of this. Our second case study indicated that context might actually be secondary to other aspects, such as desire, in shaping the encounter as spiritually authorizing.

## 6. Case Study: Spirituality Chatbot

The question of AI as "spiritually authorizing" outside of traditional religious contexts was made acutely relevant in the work of Lu Xiao and the Spiritual Chatbot Project (Asante-Agyei et al. 2022). Xiao and her team conducted an interview study to determine the willingness and hesitancy of participants to engage in conversations concerning religion and spirituality with a chatbot.[5] Their pilot study used Juji Studio, an online platform that allows users to design, build, and launch the conversation flow and persona of a chatbot. The researchers were looking to answer the questions: "How do people perceive the use of chatbots for spiritual purposes?" and "What preferences do people have for chatbots for spiritual purposes, if any?".

In contrast to other chatbots associated with religion, Xiao and her team have taken a unique bottom-up approach that tracks user comments and suggestions to gauge interest in a "spiritual chatbot", rather than assuming what users want (for instance, Pataranutaporn et al. 2019). The initial conclusions of the study pointed not only to a willingness to engage an AI as part of a religious practice, but also the desire for an AI to provide responses to questions and provide affirmations of religious identity, belief, and practice. The findings of the study contrasted one track of cultural assumptions about the use of AI for religious and spiritual formation—namely, the Western repertoire of nontheistic (cf. Singler 2020), malevolent, *Terminator*-inspired visions of AI as dangerous (Singler 2019). Specifically, study participants seemed to desire the AI to be more authoritative in terms of spiritual direction. For example, participants wanted the Chatbot to be "more spiritual", lead them through prayers, prompt them to reflect, and answer questions about the histories and meanings behind different religious persons and practices.[6] Xiao and her team drew two major conclusions from these findings.

One conclusion of Xiao et al.'s study involved the user's agency in choosing the visual display of the Chatbot they encountered. Part of the strategy of the interview team was to give interviewees a choice over which "avatar" and persona they wanted their Chatbot to inhabit. Those who chose the robot (Juji) tended to expect more informational engagements, while those who chose the human (Ava) expected more empathy and understanding. This avatar selection option fell in line with other studies (not referenced by Xiao et al.) that proved that giving users agency over their own avatar choice could increase immersive engagement in digital–virtual experiments (See Harrell 2013).[7]

The other conclusion was more interesting for our exploration in this paper. It was that participants had expectations that a spirituality chatbot should offer cognitive empathy (not necessarily emotional empathy) as much as it should offer content expertise. From their responses, participants' concern regarding a spirituality chatbot being empathetic referred to the cognitive empathy that involves a capacity for understanding the other's emotional state and acting in a comforting and appropriate way in the situation. Users indicated that their desire for a spiritually authorizing encounter could be satisfied through

the AI's language; specifically, its performance of a type of reflective listening.[8] Here, the participants seemed to shift the valuative problem from visual displays of emotion or "humanness" that many AI and robot studies link with uncanniness, toward the use of reflective empathy in language. There was similarity here to the ELIZA effect discussed above, highlighting the function of linguistic empathy, rather than merely dialogical interest (the fact of responding in a conversational, turn-taking manner). The ontology of the AI's "agency" or personhood was less important than its behavior when it came to grounding spiritually or religiously authorizing encounters. Yet, what the AI is, materially speaking, will be important for the kinds of encounters it can ground as the makers of machine agents such as the Spirituality Chatbot and Mindar seek to grow them in complexity.

The neuroscientist Anil Seth made the argument (e.g., Seth 2021; Seth and Tsakiris 2018) that AI will never be conscious in the same way a human is, nor will it have the kind of interior mental faculties as even our distant mammalian relatives, because our mind|brains evolved through a process of homeostatically regulating the biological bodies of which they were a part, and this is simply not a function that machine bodies need to fulfill. There is a parallel and mirror move in Taves, regarding the way that humans singularize and attribute agency. That is, one of Taves' valuable additions to the study of experiences deemed religious or special in *RER* and after was to place human tendencies to set things apart in a wider mammalian context. That is, she highlighted how drawing boundaries—for example, around what one does and does not do with kin—is an evolved animal tendency. Crucially, as Gustavo Benavides (2010) highlighted in a friendly critique of *RER*, the way things are set apart as special in both positive (e.g., the mother–infant bond) and negative ways (e.g., incest taboos) is done through sensory cues of smell, taste, touch, sight, sound, etc. Of course, machine bodies are sensitive in their own ways to patterns of matter|energy, but not to the ones, or not in the ways, that our homeostatic bodies have evolved to sense. Thus, in the case of encounters with Mindar or the Spirituality Chatbot, it is as if the building block of simple ascription is tentatively in play, insofar as fabulation is an ever-present tendency of the human body, and the building block of singularization is suggested by the fact of these machine agents being contextualized in relation to special paths, but it remains largely in reserve for further cues as to whether and how the machine agent should be set apart as special in some way.

## 7. Conclusions

Rethinking the case studies above, we contend that users encountered the behavior of these machine agents as agential, and deem it to be spiritually authorizing, as a result of the users' fabulative capacity for perceiving agency. This agency took on a spiritual or religious valence when contextualized as such for the user (which is why people who experience the ELIZA effect do not necessarily valence the encounter as religious) in both the temple setting (Mindar) and the framing of the experiment (Spiritual Chatbot). What fabulation reminds us, however, is that this perception of agency is also shaped by individuated affective changes within the perceiver's body. In other words, the affective capacities of the body are valenced by religious contexts that trigger conscious and unconscious experiences of our own agency, which then allow for transference onto external agents.

To conclude, our perceptions of machine agents such as robots and AI as being spiritual or religious in a way that authorizes behavioral changes involves processes and tendencies of the human body more so than it does any metaphysical nature of the technology itself. The designers who realize this will be able to code for more affective experiences (for good and for ill), while the users who are unaware are likely to be more affected. Our encounters with machine agents, then, must come with a more precise skepticism. To be clear, these encounters with machine agents are not less "real" than encounters with other spiritual authorizers (human religious leaders). Rather, our bodies play a role in shaping the reality of these encounters regardless of the agency involved (human or nonhuman).

Given the rapid pace of the development of both AI and robotics, qualitative and quantitative studies are playing perpetual catch-up. Crucially, we must recognize that the

margin of error in popular judgments about the advanced state of AI and robotics (and the other "GRIN" technologies) is always shifting, and at least in some respects, gradually closing as computing and robotics do indeed advance. Therefore, it is vital to continually reconsider how people adjudicate where the boundary lies between AI cognition and human mind | brains, the natural and the machine, and between matter and spirit. Our work here aimed to contribute to this task by showing the role of embodied tendencies of fabulation, singularization, and simple ascription in shaping attitudes and experiences of both existing and speculative technologies, so that we might better anticipate, understand, and design for the kind of religious, spiritual, and special relationships that will be possible.

**Author Contributions:** Conceptualization; methodology; resources; writing—original draft preparation, review and editing, J.L.-C. and S.C.M.; funding acquisition, S.C.M. All authors have read and agreed to the published version of the manuscript.

**Funding:** This research was funded by New Frontiers in Research Fund grant number NFRFE-2019-01542. And The APC was waived.

**Institutional Review Board Statement:** Not applicable.

**Informed Consent Statement:** Not applicable.

**Data Availability Statement:** Not applicable.

**Acknowledgments:** We would like to acknowledge and thank our Co-PI Amber Simpson.

**Conflicts of Interest:** The authors declare no conflict of interest.

## Notes

[1] Following Isabel Millar (2021), we prefer "machine agent" rather than "artificial intelligence" to resist the ways that "intelligence" has been fetishized over and against other types of machine–human-agential action. Nevertheless, we continue to use AI as a connotative shorthand.

[2] For an excellent deflation of this kind of apocalyptic thinking, see Alenka Zupančič (2018), Discussing "the 'apocalyptic mood' in recent times" (16), Zupančič exposes how "its actually taking place can never match up to the power it wields in being withheld—that is, with its remaining a threat" (17). Also, for a valuable approach to thinking apocalypse with psychoanalytic concepts from a religious studies perspective, see Dustin Atlas's (Forthcoming).

[3] Some critiques of Taves' building-block approach have highlighted how this framework better describes originary and phenomenologically extraordinary moments in the "special paths" that Taves identifies with those things often deemed "religions" and "spiritualities" than it does the ordinary, habitual moments of comparatively unreflexive religious and spiritual practice that take place once these special paths get taken up by others over time. For instance, Kim Knott (2010, p. 305) writes: "My criticism is simply that Taves does not give sufficient attention to other work arising from the same Durkheimian source which precedes and potentially informs her own project, work that has focused on and developed the concept of the 'sacred', arguably with similar intentions and an equal commitment to resisting a sui generis interpretation of religion and of the sacred itself. . . . [Notably] Durkheim's articulation of the 'sacred' as that which is set apart from the profane or ordinary and protected by taboos, not least of all his assessment that 'anything can be sacred.'" The building-block approach's focus on the novel actually suits our application here to machine agency insofar as machine agency is at this historical moment only familiar within some philosophical circles, and insofar as AI and digital robotics are historically recent and rapidly advancing domains.

[4] See (Hayles 1999).

[5] Xiao's team writes, "we acknowledge the distinction between spirituality and religion, though it is beyond the discussion of this study. Currently, this study focuses on spiritual activities in religious contexts" (1). While the study authors took a bottom-up approach in discovering what people want from a chatbot, they took a top-down approach in assuming that there is a transcontextual core of elements in particular religions that constitutes the spiritual. Their study data showed that participants' own emic understandings of spirituality would have been a more grounded place to start for a thoroughly bottom-up approach to the study design.

[6] To be clear, while their findings are compelling, there are a host of problems with the construction and implementation of the study itself beyond the fact that it incompletely theorizes both "spirituality" and "religion". To name a few, the study makes implicit theological assumptions, assumes a Perennialism, and limits its data set to "Bible" and "Qu'ran" Reddit boards only constructing the chatbot responses.

[7] In "religious" terms, they can choose to make the bot (religious authority) in their own image.

[8] For a clarification of Rogerian reflective listening, in particular, see Kyle Arnold (2014).

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
