# Peer review of "Fabulation, Machine Agents, and Spiritually Authorizing Encounters"

_religions, doi:10.3390/rel13040333_

Round 1

Reviewer 1 Report

The article is very clear in its examples and extremely helpful for the debate on AI. However, I do not see any reason for the theoretical frame of Bergson (ch. 4) except perhaps for some unnecessary explanation of the term "fabulation". I would therefore suggest to delete ch. 4 and the reference to it in the beginning, and this moreover because Bergson's ideas do not show up in the Conclusion.

Author Response

We appreciate reviewer 1’s response and have taken it into account. Since Bergson’s particular conceptualization of fabulation is key to this unique (albeit modest) reading of AI, rather than removing Bergson entirely, we have attempted to highlight and carry through his unique take by coupling it with the theories of Anne Taves in order to provide a fuller reading of the Bergsonian concept throughout. We also added more of the fabulative reading to the conclusion so that it is carried as a through-line in the paper. Please see lines 343, 347, 370. 

Reviewer 2 Report

Fine article, well structured. Some small typographical editing necessary (including some block quotations that are not indented, line 181), some incomplete editing (line 237), some lack of editorial parallelism ( (Juji) vs "Ava") line 261-262); line 269: users'

I do wonder if some more explicit engagement with the term "special" is necessary -- and possibly consideration if this is the best term if it always needs to be in scare quotes. As this relates directly to the conclusion argued at the end, it might be worth adding a few more paragraphs considering what determines human experience as "religiously 'special'" (or whatever term you end up using) and what differentiates that from both other kinds of heightened (?) experiences and from everyday life. 

Author Response

We have made all the typographical edits suggested by the reviewer. Thank you for catching them! We also appreciated the suggestion to expand on the reading of “special.” What is nice, is that, to date, no one yet has interpreted human relations with AI through Taves’ building blocks approach, and since it is fruitful to do so, we will make an initial attempt here. In line with the suggestion for expansion, we added an entire extra section focusing on Taves’ building block approach (see page 5), and we also included a footnote (footnote #3) referencing Kim Knott’s critique of Taves distinction between “sacred,” “special,” and “religious” in order to capture the importance of context in shaping the deeming of an experience specifically as “religious."